# The Influence of CB2-Receptor Ligands on the Memory-Related Responses in Connection with Cholinergic Pathways in Mice in the Passive Avoidance Test

**DOI:** 10.3390/molecules27134252

**Published:** 2022-07-01

**Authors:** Marta Kruk-Slomka, Agnieszka Dzik, Grazyna Biala

**Affiliations:** Department of Pharmacology and Pharmacodynamics, Medical University of Lublin, 4a Chodzki Str., 20-093 Lublin, Poland; agnieszkadzik2016@gmail.com (A.D.); grazyna.biala@umlub.pl (G.B.)

**Keywords:** cholinergic pathways, cognitive disorders, memory and learning, CB2 receptor ligands, mice, passive avoidance test

## Abstract

**Background:** Dysfunction of the cholinergic system is associated with the development of Alzheimer’s disease (AD). One of the new possible strategies for the pharmacological modulation of memory-related problems typical of AD, is connected with the endocannabinoid system (ECS) and the cannabinoid (CB: CB1 and CB2) receptors. **Methods:** The aim of the study was to determine the influence of the selective CB2 receptor ligands: agonist (JWH 133) and antagonist (AM 630) on different stages of memory and learning in mice, in the context of their interaction with cholinergic pathways. To assess and understand the memory-related effects in mice we used the passive avoidance (PA) test. **Results:** We revealed that co-administration of non-effective dose of JWH 133 (0.25 mg) or AM 630 (0.25 mg/kg) with the non-effective dose of cholinergic receptor agonist - nicotine (0.05 mg/kg) enhanced cognition in the PA test in mice; however, an acute injection of JWH 133 (0.25 mg/kg) or AM 630 (0.25 mg/kg) had no influence on memory enhancement induced by the effective dose of nicotine (0.1 mg/kg). Co-administration of JWH 133 (0.25 mg) or AM 630 (0.25 mg/kg) with the effective dose of the cholinergic receptor antagonist scopolamine (1 mg/kg) attenuated the scopolamine-induced memory impairment in the PA test in mice. **Conclusion:** Our experiments have shown that CB2 receptors participate in the modulation of memory-related responses, especially those in which cholinergic pathways are implicated.

## 1. Introduction

Dysfunction of cholinergic neurons is responsible for the development of many central nervous system (CNS) diseases, including Alzheimer’s disease (AD). In the course of AD, degeneration of cholinergic neurons occurs, which leads to a reduction in acetylcholine (ACh) level, a neurotransmitter that is crucial for memory and learning processes. Additionally, AD pathogenesis is widely believed to be driven by the production and deposition of the β-amyloid peptide (Aβ). 

As a result of these changes, impairment of cholinergic transmission follows, which then results in typical symptoms of this disease [1]. Dysfunctions related to the formation of memory pathways are the main symptoms of AD, however, no less significant disturbances in the emotional sphere of equal if not greater severity are also observed, i.e., anxiety, depression, psychosis, often appearing as a consequence of memory deficits [2]. The decrease in the concentration of the ACh, still remains the main target of symptomatic treatment of AD (cholinergic strategy) [3]. Unfortunately, this therapy does not regulate the emotional state of the patient, so it does not alleviate the emotional symptoms common in AD [4], and, moreover, can evoke depression in the patient by increasing the levels of ACh in the central nervous system (CNS), as a side effect of mainstream AD therapy [2]. Therefore, there is a justified need to thoroughly understand the mechanisms determining the development of AD, and which are thus responsible for the symptoms of this disease, in order to facilitate the search for new strategies that could potentially be used in modulating the course of AD in the future.

High hopes for the treatment and control of AD symptoms are associated with the endocannabinoid system (ECS). The ECS, through the cannabinoid type 1 (CB1) and type 2 (CB2) receptors, is involved in many physiological functions, e.g., in memory and learning processes; however, most research in this context concerns the potential role of CB1 receptor [5,6,7,8,9,10], while the specific impact of CB2 receptor ligands on the cognition-related processes seems to be more complex and still not precisely explored; moreover, taking into account the fundamental role of the cholinergic system in the pathogenesis of AD, it seems extremely important to capture the interactions between ECS and the cholinergic system.

Following that, the aim of the presented study was to determine for the first time the influence of the selective CB2 receptor ligands on the different stages of memory processes in the context of the interactions with the cholinergic system. We examined the impact of CB2 receptor agonist (JWH 133) and antagonist (AM 630) on the memory acquisition and consolidation processes modified by an acute administration of nicotine, a cholinergic receptor agonist, as well as scopolamine, a cholinergic receptor antagonist, in mice. To assess memory-related effects in mice we used the passive avoidance (PA) test. PA is a commonly used behavioral test that allows evaluating different stages of memory depending on the drug treatment. 

These innovative studies allow for a deeper assessment of the interaction of the ECS and the cholinergic system in the context of memory and learning processes. Our results will be discussed especially in the connection with the influence of CB2 receptor ligands on the memory-related processes connected with cholinergic pathways. Due to the specific impact of CB2 receptor ligands on the cognition-related processes and their pharmacological properties, these cannabinoid compounds appear to be a promising therapeutic target in the treatment of memory-related disorders, especially those in which cholinergic pathways are implicated. As such, our findings could be helpful in further research concerning the effective pharmacotherapy of the diseases which are associated with cognitive impairments, especially those referred to as cholinergic system dysfunctions, such as AD.

## 2. Results

### 2.1. The Impact of Cholinergic Ligands on the Memory and Learning Processes in Mice in the PA Test

In the first step of our experiments, we confirmed the opposite effects of the acute injection of cholinergic receptor ligands: nicotine and scopolamine on the memory and learning processes in mice using the PA test. 

One-way ANOVA revealed that the administration of acute doses of cholinergic receptor ligands had a statistically significant effect on LI values for long-term memory acquisition [F(3.34) = 22.59; *p* < 0.0001], as well as long-term memory consolidation [F(3.35) = 10.84; *p* < 0.0001].

In the case of memory acquisition, a Tukey’s post hoc test confirmed that an acute s.c. administration of nicotine at the higher dose (0.1 mg/kg) significantly increased LI values in mice compared to those in the vehicle-treated control group (*p* < 0.05) indicating that nicotine, at this used dose, improved the long-term acquisition of memory and learning processes in the PA test in mice. In turn, an acute i.p. administration of scopolamine (1 mg/kg) significantly decreased LI values in mice compared to those in the vehicle-treated control group (*p* < 0.001; Tukey’s test) indicating that scopolamine, at this used dose, impaired the long-term acquisition of memory and learning processes in the PA test in mice (Figure 1A).

Similarly, in the case of consolidation of memory, a Tukey’s post hoc test confirmed that an acute s.c. administration of nicotine at the higher dose (0.1 mg/kg) significantly increased LI values in mice compared to those in the vehicle-treated control group (*p* < 0.01) indicating that nicotine, at this used dose, improved the long-term consolidation of memory and learning processes in the PA test in mice. An acute i.p. administration of scopolamine (1 mg/kg) significantly decreased LI values in mice compared to those in the vehicle-treated control group (*p* < 0.05) indicating that scopolamine, at this used dose, impaired the long-term consolidation of memory and learning processes in the PA test in mice (Figure 1B).

### 2.2. The Impact of CB2 Receptor Ligands on the Memory and Learning Processes in Mice in the PA Test

In the next step of our experiments, we revealed the positive influence of the acute injection of CB2 receptor ligands on the memory and learning processes in mice using the PA test. 

#### 2.2.1. The Impact of the CB2-Receptor Agonist JWH 133 on the Memory and Learning Processes in Mice in the PA Test

One-way ANOVA revealed that the administration of acute i.p. doses of CB2 receptor agonist JWH 133 had no statistically significant effect on the LI values for long-term memory acquisition [(F(3.36) = 0.7008; *p* = 0.5537]. Results indicated that JWH 133 (0.25; 0.5; 1 mg/kg), at these used doses had no influence on the long-term acquisition of memory and learning processes in the PA test in mice (Figure 2A).

In the case of memory consolidation, one-way ANOVA revealed that the administration of acute i.p. doses of JWH 133 had a statistically significant effect on LI values for long-term memory acquisition [F(3.33) = 9.384; *p* = 0.0002]. Additionally, a Tukey’s *post hoc* test confirmed that an acute i.p. administration of JWH 133 at the higher doses (0.5 and 1 mg/kg) significantly increased LI values in mice compared to those in the vehicle-treated control group (*p* < 0.001 for a dose of 0.5 mg/kg and *p* < 0.01 for a dose of 1 mg/kg) indicating that JWH 133, at these used doses, improved the long-term consolidation of memory and learning processes in the PA test in mice (Figure 2B).

#### 2.2.2. The Impact of the CB2-Receptor Antagonist AM 630 on the Memory and Learning Processes in Mice in the PA Test

One-way ANOVA revealed that the administration of acute i.p. doses of the CB2-receptor antagonist AM 630 (0.25; 0.5; 1 mg/kg) had statistically significant effect on the LI values for long-term memory acquisition (F(3.35)= 6.946; *p* = 0.0010), as well as for long-term memory consolidation (F(3.29)= 5.504; *p* = 0.0046) in the PA test in mice.

In the case of memory acquisition, a Tukey’s post hoc test confirmed that an acute i.p. administration of AM 630 at the higher doses (0.5 and 1 mg/kg) significantly increased LI values in mice compared to those in the vehicle-treated control group (*p* < 0.05 for a dose of 0.5 mg/kg and *p* < 0.001 for a dose of 1 mg/kg) indicating that AM 630, at these used doses, improved the long-term acquisition of memory and learning processes in the PA test in mice (Figure 3A). 

Similarly, in the case of consolidation of memory, a Tukey’s post hoc test confirmed that an acute i.p. administration of AM 630 at the higher doses (0.5 and 1 mg/kg) significantly increased LI values in mice compared to those in the vehicle-treated control group (*p* < 0.05 for a dose of 0.5 mg/kg and *p* < 0.01 for a dose of 1 mg/kg) indicating that AM 630, at these used doses, improved the long-term consolidation of memory and learning processes in the PA test in mice (Figure 3B).

Based on the results obtained from these two experiments in the PA task, the non-effective (0.05 mg/kg) and effective (0.1 mg/kg) doses of nicotine, and effective dose of scopolamine (1 mg/kg) were then chosen for the next behavioral experiment evaluating the involvement of CB2 receptor ligands (JWH 133 at the non-effective dose of 0.25 mg/kg; AM 630 at the non-effective dose of 0.25 mg/kg) in memory-related behavior provoked by cholinergic ligands, described below.

### 2.3. The Influence of the Administration of CB2 Receptor Ligands on the Memory-Related Responses Provoked by Nicotine in the PA Test in Mice

#### 2.3.1. The Impact of the CB2 Receptor Agonist JWH 133 on the Memory and Learning Processes Provoked by Non-Effective Dose of Nicotine in Mice in the PA Test

For memory acquisition, two-way ANOVA analyses revealed that there was a statistically significant effect caused by nicotine (0.05 mg/kg) treatment [F(1.36) = 8.107; *p* = 0.0072], JWH 133 (0.25 mg/kg) pretreatment [F(1.36) = 16.00; *p* = 0.0003] and interactions [F(1.36) = 16.45; *p* = 0.0003]. Additionally, *post-hoc* Bonferroni’s test revealed that co-administration of JWH 133 (0.25 mg/kg) with the non-effective dose of nicotine (0.05 mg/kg) significantly increased LI values in mice in the PA test in comparison to: (i) the vehicle/vehicle-treated mice (*p* < 0.001); (ii) vehicle/nicotine (0.05 mg/kg)-treated mice (*p* < 0.001); (iii) JWH 133 (0.25 mg/kg)/vehicle-treated mice (*p* < 0.001) confirming that co-administration of JWH 133 and nicotine at non-effective doses enhanced acquisition of cognition in the PA test in mice (Figure 4A). 

For memory consolidation, two-way ANOVA analyses revealed that there was no statistically significant effect caused by nicotine (0.05 mg/kg) treatment [F(1.31) = 1.720; *p* = 0.1994], but there was a statistically significant effect caused by JWH 133 (0.25 mg/kg) pretreatment [F(1.31) = 9.405; *p* = 0.0045] and interactions [F(1.31) = 5.951; *p* = 0.0206]. Additionally, post-hoc Bonferroni’s test revealed that co-administration of JWH 133 (0.25 mg/kg) with the non-effective dose of nicotine (0.05 mg/kg) significantly increased LI values in mice in the PA test in comparison to: (i) the vehicle/vehicle-treated mice (*p* < 0.01); (ii) vehicle/nicotine (0.05 mg/kg)-treated mice (*p* < 0.001); (iii) JWH 133 (0.25 mg/kg)/vehicle-treated mice (*p* < 0.05) confirming that co-administration of JWH 133 and nicotine at non-effective doses enhanced consolidation of cognition in the PA test in mice (Figure 4B). 

Non-effective doses of the CB2 receptor agonist (JWH 13, 0.25 mg/kg) or vehicle were administered 15 min prior to a non-effective dose of cholinergic receptor agonist (nicotine, 0.05 mg/kg). Injections were performed 15 min before the first trial (acquisition of memory) (A) or immediately after the first trial (consolidation of memory) (B). 24 h later, the second trial was conducted; *n* = 8–12; the means ± SEM; *** *p* < 0.001; ** *p* < 0.01 vs. vehicle/vehicle-treated group; ^^^ *p* < 0.001 vs. vehicle/nicotine (0.05 mg/kg)-treated group; ### *p* < 0.001; # *p* < 0.05 vs. JWH 133 (0.25 mg/kg)/vehicle-treated group; Bonferroni’s test.

#### 2.3.2. The Impact of the CB2 Receptor Antagonist AM 630 on the Memory and Learning Processes Provoked by a Non-Effective Dose of Nicotine in Mice in the PA Test

For memory acquisition, two-way ANOVA analyses revealed that there was no statistically significant effect caused by nicotine (0.05 mg/kg) treatment [F(1.33) = 0.4890; *p* = 0.4893] and interactions [F(1.33) = 0.0643; *p* = 0.8012], but there was statistically significant effect by AM 630 (0.25 mg/kg) pretreatment [F(1.33) = 9.149; *p* = 0.0048] (Figure 5A). 

In turn, for memory consolidation, two-way ANOVA analyses revealed that there was no statistically significant effect caused by nicotine (0.05 mg/kg) treatment [F(1.27) = 1.912; *p* = 0.1781], but there was statistically significant effect caused by AM 630 (0.25 mg/kg) pretreatment [F(1.27) = 8.682; *p* = 0.0065] and interactions [F(1.27) = 5.256; *p* = 0.0299]. Additionally, post hoc Bonferroni’s test revealed that co-administration of AM 630 (0.25 mg/kg) with the non-effective dose of nicotine (0.05 mg/kg) significantly increased LI values in mice in the PA test in comparison to: (i) vehicle/vehicle-treated mice (*p* < 0.05); (ii) vehicle/nicotine (0.05 mg/kg)-treated mice (*p* < 0.01); (iii) AM 630 (0.25 mg/kg)/vehicle-treated mice (*p* < 0.05) confirming that co-administration of AM 630 and nicotine at non-effective doses enhanced consolidation of cognition in the PA test in mice (Figure 5B). 

Non-effective doses of the CB2 receptor antagonist (AM 630, 0.25 mg/kg) or vehicle were administered 15 min prior to non-effective dose of cholinergic receptor agonist (nicotine, 0.05 mg/kg). Injections were performed 15 min before the first trial (acquisition of memory) (A) or immediately after the first trial (consolidation of memory) (B). 24 h later the second trial was conducted; *n* = 8–12; the means ± SEM; * *p* < 0.05 vs. vehicle/vehicle-treated group; ^^ *p* < 0.01 vs. vehicle/nicotine (0.05 mg/kg)-treated group; # *p* < 0.05 vs. AM 630 (0.25 mg/kg)/vehicle-treated group; Bonferroni’s test.

#### 2.3.3. The Impact of the CB2 Receptor Agonist JWH 133 on the Memory and Learning Processes Provoked by Effective (Pro-Cognitive) Dose of Nicotine in Mice in the PA Test

For memory acquisition, two-way ANOVA analyses revealed that there was a statistically significant effect caused by nicotine (0.1 mg/kg) treatment [F(1.31) = 17.05; *p* = 0.0003] but there was no statistically significant effect caused by JWH 133 (0.25 mg/kg) pretreatment [F(1.31) = 0.8149; *p* = 0.3736], and interactions [F(1.31) = 0.9272; *p* = 0.3430]. Additionally, post hoc Bonferroni’s test confirmed that an acute administration of nicotine (0.1 mg/kg) significantly increased LI values in mice in the PA test in comparison to the vehicle/vehicle-treated mice (*p* < 0.05) confirming that nicotine at this dose used enhanced acquisition of cognition in the PA test in mice (Figure 6A). 

Similarly, for memory consolidation, two-way ANOVA analyses revealed that there was statistically significant effect caused by nicotine (0.1 mg/kg) treatment [F(1.31) = 8.370; *p* = 0.0069] but there was no statistically significant effect caused by JWH 133 (0.25 mg/kg) pretreatment (F(1.31) = 0.2037; *p* = 0.6549), and interactions (F(1.31) = 0.001346; *p* = 0.9710). Additionally, post hoc Bonferroni’s test confirmed that an acute administration of nicotine (0.1 mg/kg) significantly increased LI values in mice in the PA test in comparison to the vehicle/vehicle-treated mice (*p* < 0.05) confirming that nicotine at this dose used enhanced acquisition of cognition in the PA test in mice (Figure 6B). 

Non-effective doses of the CB2 receptor agonist (JWH 133, 0.25 mg/kg) or vehicle were administered 15 min prior to the pro-cognitive dose of cholinergic receptor agonist (nicotine, 0.1 mg/kg). Injections were performed 15 min before the first trial (acquisition of memory) (A) or immediately after the first trial (consolidation of memory) (B); 24 h later, the second trial was conducted; *n* = 8–12; the means ± SEM; * *p* < 0.05 vs. vehicle/vehicle-treated mice; Bonferroni’s test.

#### 2.3.4. The Impact of the CB2-Receptor Antagonist AM 630 on the Memory and Learning Processes Induced by Effective (Pro-Cognitive) Dose of Nicotine in Mice in the PA Test

For memory acquisition, two-way ANOVA analyses revealed that there was statistically significant effect caused by nicotine (0.1 mg/kg) treatment (F(1.28) = 0.6830; *p* = 0.0143) but there was no statistically significant effect caused by AM 630 (0.25 mg/kg) pretreatment [F(1.28) = 2.038; *p* = 0.1645], and interactions [F(1.28) = 0.0213; *p* = 0.8850]. Additionally, post hoc Bonferroni’s test confirmed that an acute administration of nicotine (0.1 mg/kg) significantly increased LI values in mice in the PA test in comparison to the vehicle/vehicle-treated mice (*p* < 0.05) confirming that nicotine at this dose used enhanced acquisition of cognition in the PA test in mice (Figure 7A). 

Similarly, for memory consolidation, two-way ANOVA analyses revealed that there was statistically significant effect caused by nicotine (0.1 mg/kg) treatment (F(1.30) = 7.422; *p* = 0.00106) but there was no statistically significant effect caused by AM 630 (0.25 mg/kg) pretreatment (F(1.30) = 0.005963; *p* = 0.9390), and interactions (F(1.30) = 1.672; *p* = 0.2059). Additionally, post hoc Bonferroni’s test confirmed that an acute administration of nicotine (0.1 mg/kg) significantly increased LI values in mice in the PA test in comparison to the vehicle/vehicle-treated mice (*p* < 0.05) confirming that nicotine at this dose used enhanced acquisition of cognition in the PA test in mice (Figure 7B).

Non-effective dose of CB2-receptor antagonist (AM 630, 0.25 mg/kg) or vehicle were administered 15 min prior to pro-cognitive dose of cholinergic receptor agonist (nicotine, 0.1 mg/kg). Injections were performed 15 min before the first trial (acquisition of memory) (A) or immediately after the first trial (consolidation of memory) (B). 24 h later the second trial was conducted; *n* = 8–12; the means ± SEM; * *p* < 0.05 vs. vehicle/vehicle-treated mice; Bonferroni’s test.

### 2.4. The Influence of the Administration of CB2-Receptor Ligands on the Memory-Related Responses Provoked by Scopolamine in the PA Test in Mice

#### 2.4.1. The Impact of the CB2-Receptor Agonist JWH 133 on the Memory and Learning Processes Induced by Effective (Amnestic) Dose of Scopolamine in Mice in the PA Test

For memory acquisition, two-way ANOVA analyses revealed that there was statistically significant effect caused by scopolamine (1 mg/kg) treatment [F(1.34) = 2.63; *p* < 0.0001] and caused by JWH 133 (0.25 mg/kg) pretreatment (F(1.34) = 7.715; *p* = 0.0088) and interactions (F(1.34) = 8.125; *p* = 0.0074). Additionally, a post hoc Bonferroni’s test confirmed that an acute administration of scopolamine (1 mg/kg) significantly decreased LI values in mice in the PA test in comparison to the vehicle/vehicle-treated mice (*p* < 0.001) confirming that scopolamine at this dose used impaired acquisition of cognition in the PA test in mice; this amnestic effect of scopolamine (1 mg/kg) was blocked by JWH 133 (0.25 mg/kg) (*p* < 0.01 vs. vehicle/scopolamine (1 mg/kg)-treated mice) [Figure 8A].

Similarly, for memory consolidation, two-way ANOVA analyses revealed that there was statistically significant effect caused by scopolamine (1 mg/kg) treatment (F(1.28) = 4.247; *p* = 0.0487) and caused by JWH 133 (0.25 mg/kg) pretreatment (F(1.28) = 20.21; *p* = 0.0001) and interactions (F(1.28) = 7.713; *p* = 0.0097). Additionally, a post hoc Bonferroni’s test confirmed that an acute administration of scopolamine (1 mg/kg) significantly decreased LI values in mice in the PA test in comparison to the vehicle/vehicle-treated mice (*p* < 0.05) confirming that scopolamine at this dose used impaired consolidation of cognition in the PA test in mice; this amnestic effect of scopolamine (1 mg/kg) was blocked by JWH 133 (0.25 mg/kg) (*p* < 0.001 vs. vehicle/scopolamine (1 mg/kg)-treated mice) [Figure 8B].

Non-effective dose of CB2-Receptor agonist (JWH 133, 0.25 mg/kg) or vehicle were administered 15 min prior to amnestic dose of cholinergic receptor antagonist (scopolamine, 1 mg/kg). Injections were performed 15 min before the first trial (acquisition of memory) (A) or immediately after the first trial (consolidation of memory) (B). 24 h later, the second trial was conducted; *n* = 8–12; the means ± SEM; * *p* < 0.05; *** *p* < 0.001 vs. vehicle/vehicle-treated group; ^^ *p* < 0.001; ^^^ *p* < 0.001 vs. vehicle/scopolamine (1 mg/kg)-treated group; Bonferroni’s test.

#### 2.4.2. The Impact of the CB2-Receptor Antagonist AM 630 on the Memory and Learning Processes Induced by Effective (Amnestic) Dose of Scopolamine in Mice in the PA Test

For memory acquisition, two-way ANOVA analyses revealed that there was statistically significant effect caused by scopolamine (1 mg/kg) treatment [F(1.33) = 45.51; *p* < 0.0001] and caused by AM 630 (0.25 mg/kg) pretreatment [F(1.33) = 21.19; *p* < 0.0001] and interactions [F(1.33) = 5.650 *p* = 0.0234]. Additionally, a post hoc Bonferroni’s test confirmed that acute administration of scopolamine (1 mg/kg) significantly decreased LI values in mice in the PA test in comparison to the vehicle/vehicle-treated mice (*p* < 0.001) confirming that scopolamine at this dose used impaired acquisition of cognition in the PA test in mice; this amnestic effect of scopolamine (1 mg/kg) was blocked by AM 630 (0.25 mg/kg) (*p* < 0.001 vs. vehicle/scopolamine (1 mg/kg)-treated mice). Additionally, post hoc Bonferroni’s test revealed that co-administration of AM 630 (0.25 mg/kg) with the effective (amnestic) dose of scopolamine (1 mg/kg) significantly increased LI values in mice in the PA test in comparison to AM 630 (0.25 mg/kg)/vehicle-treated mice (*p* < 0.05) [Figure 9A].

For memory acquisition, two-way ANOVA analyses revealed that there was statistically significant effect caused by scopolamine (1 mg/kg) treatment (F(1.30) = 12.06; *p* = 0.0016) and caused by AM 630 (0.25 mg/kg) pretreatment [F(1.30) = 24.73; *p* < 0.0001] and interactions [F(1.30) = 23.21; *p* < 0.0001]. Additionally, post hoc Bonferroni’s test confirmed that an acute administration of scopolamine (1 mg/kg) significantly decreased LI values in mice in the PA test in comparison to the vehicle/vehicle-treated mice (*p* < 0.001) indicating that scopolamine at this dose used impaired consolidation of cognition in the PA test in mice; this amnestic effect of scopolamine (1 mg/kg) was blocked by AM 630 (0.25 mg/kg) (*p* < 0.001 vs. vehicle/scopolamine (1 mg/kg)-treated mice) [Figure 9B].

Non-effective doses of CB2-receptor antagonist (AM 630, 0.25 mg/kg) or vehicle were administered 15 min prior to amnestic dose of cholinergic receptor antagonist (scopolamine, 1 mg/kg). Injections were performed 15 min before the first trial (acquisition of memory) (A) or immediately after the first trial (consolidation of memory) (B). 24 h later, the second trial was conducted; *n* = 8–12; the means ± SEM; *** *p* < 0.001 vs. vehicle/vehicle-treated group; ^^^ vs. vehicle/scopolamine (1 mg/kg)-treated group; ^&^
*p* < 0.05 vs. AM 630 (0.25 mg/kg)/vehicle-treated mice; Bonferroni’s test.

## 3. Discussion

At the core of the development of neurodegenerative diseases, there is irreversible damage and, consequently, accelerated death of the nerve cells. AD is an example of a disease associated with the neurodegeneration of cholinergic neurons. In the course of AD, degeneration of cholinergic neurons occurs, which leads to a reduction in ACh levels. As the effect of these changes, impairment in cholinergic transmission follows, which then results in typical symptoms of this disease [1]. Pharmacotherapy of AD focuses primarily on alleviating memory impairment and employs two strategies for this purpose: cholinergic and glutamatergic [11]. The decrease in the concentration of the neurotransmitter crucial for cognitive processes, i.e., ACh still remains the main target of symptomatic treatment of AD (cholinergic strategy). Unfortunately, this therapy does not regulate the emotional state of the patient and is connected with many side effects of mainstream AD therapy [2]. Therefore, there is a justified need to thoroughly understand the mechanisms determining the development of AD, and which are thus responsible for the symptoms of this disease, in order to facilitate the search for new strategies that could potentially be used in modulating the course of AD in the future. 

One of the new strategies for the treatment and control of AD symptoms is associated with the ECS; this system, through the CB1- and CB2-receptors, is involved in many physiological functions, e.g., memory and learning processes. Especially, the specific impact of CB2-receptor ligands on the cognition-related processes seems to be more complex and still not precisely explored; thus, CB2-receptor ligands appear to be a promising target in the treatment of memory-related disorders; however, the mechanisms involved in the correlation between the cholinergic system and the ECS, through CB2-receptor ligands, in the context of AD have not been sufficiently understood. 

The aim of the study was to determine the influence of the selective CB2-receptor ligands on the different stages of memory processes in the context of the interactions with the cholinergic system. We examined the influence of cholinergic receptor ligands (nicotinic (N) receptor agonist: nicotine and muscarinic (M) receptor antagonist: scopolamine), as well as CB2-receptor ligands (agonist: JWH 133 and antagonist: AM 630) on different stages of long-term (acquisition and consolidation) memory-related responses, and in the next step, we determined the involvement of CB2-receptors on the memory-related responses connected with cholinergic pathways modulation. To assess the cognitive functions in mice, we used the PA test, commonly used in pharmacological studies.

In the first part of the study, the influence of a single administration of the substances modulating the activity of the cholinergic system (cholinergic ligands: nicotine and scopolamine) and ECS (CB2-receptor ligands: JWH 133 and AM 630) on the processes of memory acquisition and consolidation in the PA test in mice was assessed. 

### 3.1. The Impact of Cholinergic Receptor Ligands on the Memory Acquisition and Consolidation in PA Test in Mice

In this study, it was shown that nicotine has a dose-dependent modulating effect on memory and learning processes. Our studies revealed that an acute administration of nicotine (0.1 mg/kg) significantly improved memory acquisition and consolidation in the PA test in mice. A dose of 0.05 mg/kg of nicotine was found ineffective in this study.

In turn, an acute administration of scopolamine (1 mg/kg) impaired memory acquisition and consolidation in PA tests in mice. 

Our results are in accordance with our previous and available data [12,13]. Nicotine and scopolamine are substances that modulate the functioning of the cholinergic system in an opposite way; nicotine stimulates, while scopolamine inhibits the functioning of this system. Due to the opposite mechanism of action, these compounds are often used in experimental pharmacology as the model substances in many behavioral tests, e.g., for assessing memory [12,13,14,15].

Central actions of nicotine [16,17,18] and scopolamine [19,20,21,22], in the context of the effects on the processes related to the remembering of information, have been considered many times. 

A number of scientific reports describe the complex and ambiguous effect of nicotine on memory. The results of some experiments indicate an improvement of memory, induced by a single injection of nicotine [23,24], while the others do not indicate any effect of nicotine on cognitive processes [25,26], or report its inhibitory effect on the processes of formation of memory pathways [25]. The different effects of nicotine depend on the doses used, the test involved and the duration of the training session, as well as the experimental procedure related to the test used.

Nicotine is an agonist of specific N receptors, both pre- and post-synaptically located. As a result of the stimulation of presynaptic receptors, an influx of calcium ions (Ca^2+^) into the cell takes place, which in turn activates the flow of Cl^-^ ions, which finally leads to the release of a number of neurotransmitters, e.g., ACh, dopamine (DA), gamma-aminobutyric acid (GABA), glutamate (Glu), noradrenaline (NA), serotonin (5-HT) and endogenous peptides, which are directly or indirectly involved in memory processes and cognitive functions. The significant effect of nicotine on cognitive processes is also associated with the stimulation of postsynaptically located N receptors. The α7 and the α4β2 subtypes of N receptors, which are present in the highest density within the hippocampus, which is extremely important for the formation of memory pathways, play the key role in these effects [27].

Moreover, it has been shown that, as a result of nicotine administration, the activity of the above-mentioned receptors in the hippocampus area increases [28]. Stimulation of N receptor subtypes, leads to an increase in the frequency and amplitude of mini-discharges, which in turn results in the release of stored Ca^2+^, generating a nerve impulse, and as a result, positively influencing the processes related to remembering of information [24,29]. In turn, a decrease in the density of N receptors within the hippocampus and the amygdala (another structure responsible for the cognitive effects of nicotine) is associated with a significant deterioration of memory [24]. A number of scientific publications also indicate that the cognitive effects of nicotine are related not only to its influence on the cholinergic system. The mechanism related to DA or 5-HT neurotransmission seems to be of equal importance [30,31].

In the context of the procognitive effects of nicotine, it should also be noted that nicotine has a neuroprotective effect, which seems to be extremely important from the point of view of neurodegenerative diseases with memory deficits. Nicotine protects the nerve cells from the toxic effects of βA, which, as has been mentioned several times before, plays a key role in the pathogenesis of AD [32]. 

Scopolamine induces the opposite effects compared to nicotine. Scopolamine—a non-selective M receptor antagonist—causes a very strong deterioration of the ability to memorize and short-term memory impairment, which has been clearly indicated by the results of many experiments [20,33,34,35]. Memory impairment in AD patients due to dysfunction of the cholinergic system may be correlated with the memory deficits induced by scopolamine. The scopolamine model is an amnestic experimental model that impairs cholinergic transmission [36]. This study confirmed that a single administration of scopolamine (i.p.) at a dose of 1 mg/kg induces strong attenuation of memory and learning processes in mice. The results presented, confirming the amnestic effect of scopolamine, are consistent with many literature data [23,37,38]. 

The fact that in this study the amnestic effect of scopolamine was observed both in the memory acquisition and consolidation phase is consistent with literature reports which have shown that these memory phases are definitely more sensitive to the effect of scopolamine than the restoration phase [39].

As in the case of nicotine, whose procognitive effects are mainly related to the cholinergic system, the amnestic effects of scopolamine can also be partially explained by the cholinergic mechanism. M receptor agonists have been shown to improve cognitive processes that are assessed in many animal experimental models [38]. In turn, the antagonists of M receptors, i.e., atropine or scopolamine, attenuate the processes of formation of memory pathways, which is associated with the suppression of cholinergic activity. Scientific reports indicate a direct effect of scopolamine on the reduction of ACh levels in the hippocampus. The action of scopolamine is related to the non-selective blocking of M receptors (stronger M1 than M2) in the forebrain [40,41]. Scopolamine-induced impairment in cognitive processes, is abolished by cholinesterase inhibitors (IChE), which proves that cholinergic mechanisms can be involved. IChE induce an increase in the concentration of ACh in the synaptic cleft, which in turn, as a natural agonist of M receptors, competes with scopolamine for an active site within the receptor, thus inhibiting its action [28,42,43]; it has also been proven that nicotine itself impairs the amnestic effect of scopolamine, which is related to an increase of neurotransmitters releasing by nicotine, including Ach, involved in cognitive processes and the above-described mechanism of competition with scopolamine for the binding site on the M receptor [23]. 

The literature data cited above, and the results presented in this study, confirm the involvement of the cholinergic system in the process of creating memory pathways. Therefore, animal models with induced cholinergic dysfunction may find application in the search for innovative directions in the pharmacotherapy of diseases associated with memory impairment. Scopolamine is a widely employed compound, used to induce memory disorders related directly to the blockage of functions of the cholinergic system. At the same time, compounds that have a beneficial effect on the functioning of the cholinergic system can find application in the treatment of the aforementioned diseases, and it is on this assumption that the cholinergic strategy of AD treatment is based; however, due to the complex etiology of this disease, it is possible to develop new pharmacotherapy methods that would have different target points than the drugs mentioned above. One of the possible strategies is connected with ECS. As already mentioned, many literature data indicate that the ECS, through its receptors located both in the CNS and on the periphery, plays a very important role in the processes related to the formation of memory pathways and emotional behavior [44,45,46]. The experiments are underway to discover the mechanisms underlying the functioning of the ECS and their potential use in the treatment of diseases associated with cholinergic pathways-related memory impairment, e.g., AD [9,47].

### 3.2. The Impact of CB2-Receptor Ligands on Memory Acquisition and Consolidation in the PA Test in Mice

Our experiments indicated that an acute administration of a selective CB2-receptor agonist JWH 133 (0.5 and 1 mg/kg) improved memory-related responses during the acquisition stage, but not during the consolidation stage in the PA test in mice; however, an acute injection of the CB2-receptor antagonist AM 630 (0.5 and 1 mg/kg) significantly improved both acquisition and consolidation of memory in the PA test in mice. 

As previously mentioned, the involvement of the ECS in many physiological functions of the body has been confirmed. ECS plays a key role in the processes of forming memory routes and neuroprotective mechanisms [48,49].

The influence of CB1 receptor ligands on memory and learning processes has been widely documented in the scientific literature [5,6,7,8,9,10], however, the results obtained in the above experiments still contradict each other.

It has been proven that CB1 receptor agonists attenuate cognitive processes [50,51], while antagonists of these receptors exert a beneficial effect on memory and learning [51].

As indicated by the literature data cited above and the results of the presented work, agonists of CB1 receptors induce impairment of memory processes, while antagonists of these receptors show the opposite effect. The amnestic effect is induced by the activation of CB1 receptors located mainly in the areas of the brain that control cognitive processes, i.e., the cerebral cortex and hippocampus; moreover, as a result of stimulation of CB1 receptors located presynaptically on voltage-dependent calcium channels, the level of 3’,5’-cyclic adenosine monophosphate (cAMP) is reduced, which is responsible for the regulation of many extremely important functions in the cell, e.g., cell division or cell differentiation [52,53].

It is worth paying attention to the fact that the available literature mainly describes the involvement of CB1 receptors in cognitive processes. CB2-receptors are the second type of receptors, which also appear to be important, but not fully understood, involved in cognitive mechanisms. At the same time, the results of many experiments do not unequivocally confirm the effect of CB2-receptor ligands on memory. The involvement of both agonists and antagonists of CB2-receptors is important in the context of cognitive processes in the neuroprotection and neuroinflammatory processes, which indirectly affects cognitive processes and the ability to learn, especially in the course of diseases with a neuroinflammatory background with accompanying dementia [48,54,55,56,57]; this, in turn, creates the possibility of using CB2-receptor ligands as potential drugs in neurodegenerative and autoimmune diseases [49,52,58].

The effect of CB2-receptor ligands on cognitive processes appears to be very complex and is still not fully understood. Research results suggest that activation of CB2-receptors produces different effects depending on the location in the brain [59,60,61]. Literature data indicate that selective CB2-receptor agonist JWH 133 administered at the dose of 0.5 mg/kg has no effect on the acquisition of memory processes, but improves the consolidation of long-term memory in the PA test. On the other hand, higher doses (1 mg/kg and 2 mg/kg) improve both the acquisition and consolidation of long-term memory, as assessed in the same experimental model [7]; the beneficial effect on memory consolidation was also confirmed by other researchers [60].

AM 630, in turn, is one of the best-studied CB2-receptor antagonists; it acts as an inverse agonist for both CB1 and CB2-receptors [62]. It was proven that AM 630, given at lower doses, had no effect on memory in the PA test in mice, while higher doses of AM 630 induced statistically significant improvements in the antioxidant properties of brain tissues and an improvement in long-term memory in the PA test in mice, in both the acquisition and consolidation phases [7]. On the contrary to these findings, there is also literature data showing a disabling effect of AM 630 on the memory consolidation phase of the PA test in mice lacking the gene encoding the CB2-receptor [63].

The improvement in memory-related processes, induced by both agonists and antagonists of CB2-receptors, can be explained by the pharmacological properties of the ligands used. It should be noted that AM 630 acts as an inverse agonist rather than a weak antagonist [64].

The beneficial effect of CB2-receptor ligands on cognitive processes was also confirmed in this study. The results of the presented experiments indicate that both the CB2-receptor agonist JWH 133 and the CB2-receptor antagonist AM 630 improve cognition, as assessed in the PA test.

In the presented study, JWH 133 administered at doses of 0.5 mg/kg and 1 mg/kg, has a pro-cognitive effect, which is expressed by an increase in the LI value only in the consolidation phase. On the other hand, the dose of 2 mg/kg improves processes related with memory both in the acquisition and consolidation phases. AM 630 given at doses of 1 mg/kg, 2 mg/kg and 3 mg/kg, presents pro-cognitive properties, which are expressed as an increase in the LI value, both in the phase of acquisition and memory consolidation.

It should be noted that the specific cognitive effects of CB2 ligands are very complex and still not sufficiently understood. Similarly to CB1 receptor ligands, CB2-receptor ligands are capable of both disrupting and improving cognitive processes. The varied effects may be related both to the individual pharmacological characteristics of the ligands tested and to the antioxidant properties they possess, which are exhibited by both agonists and antagonists of these receptors. As already mentioned, the ECS, through CB2-receptors, is involved in neuroprotective and neuroinflammatory processes, which may result in the improvement of cognitive processes. Literature data confirm that in the course of inflammatory processes, an increased expression of CB2-receptors takes place; there are also scientific reports indicating the anti-inflammatory effect of CB2-receptor agonists [44,52,56,57]. The anti-inflammatory effect of CB2-agonists promotes the removal of toxic beta-amyloid (βA), which constitutes one of the therapeutic strategies for causal treatment of AD. It has also been proven that CB2-receptor agonist JWH 015 stimulates the activity of macrophages in terms of the ability of these cells to remove βA deposits; moreover, it has been confirmed that the administration of even a very low dose of a CB2-agonist results in significant clearance of pathological βA deposits [65].

The literature data cited above may explain why in the present work, both the agonists and antagonists of CB2-receptors show a procognitive effect, assessed in the PA test procedure in mice.

In the second step, in order to determine the interactions between the ECS and the cholinergic system, the influence of a single administration of the above-mentioned CB2-receptor ligands on cognitive processes modified by the injection of the cholinergic system ligands: nicotine and scopolamine was investigated. 

### 3.3. The Impact of CB2-Receptor Ligands on the Memory Acquisition and Consolidation Responses Induced by Cholinergic Receptor Ligands in the PA Test in Mice

Based on the results obtained from the first experiments, inactive doses of JWH 133 and AM 630 (0.25 mg/kg) were chosen for the experiment dealing with the administration of non-effective (0.05 mg/kg) or effective (0.1) nicotine or effective (1 mg/kg) scopolamine in order to show the possible antagonist/synergist effects of CB2-receptor ligands and cholinergic receptor ligands administered together. Our studies revealed that co-administration of JWH 133 (0.25 mg) or AM 630 (0.25 mg/kg) with the non-effective dose of nicotine (0.05 mg/kg) enhanced cognition in the PA test in mice; however, an acute injection of JWH 133 (0.25 mg/kg) or AM 630 (0.25 mg/kg) had no influence on memory enhancement induced by the effective dose of nicotine (0.1 mg/kg). In turn, co-administration of JWH 133 (0.25 mg) or AM 630 (0.25 mg/kg) with the effective dose of scopolamine (1.0 mg/kg) attenuated the scopolamine-induced memory impairment in the PA test in mice.

Despite the not entirely clear relationship between the cholinergic system and ECS, the results presented in the study are consistent with scientific reports that describe the correlation between CB1 and cholinergic receptors in experimental animal models. It should be noted that as a result of the activation of N receptors, induced by chronic administration of nicotine, the concentration of endogenous cannabinoids in the brain increases, e.g., in limbic structures of the forebrain [41,66,67]; there are no scientific reports concerning the effect of a single administration of nicotine on the level of endocannabinoids in the brain, although the available research results may suggest that a single injection of nicotine increases their levels in the brain structures involved in the regulation of emotional behavior [66,67]. The results of scientific studies also indicate the existence of a relationship between nicotine and CB1 receptor agonists at the biochemical level. It has been proven that combined administration of nicotine and Δ9-THC causes increased activity of the c-fos protein in such regions of the brain as PFC or the amygdala [68,69].

There is little literature data describing the relationship between the compounds modulating the functioning of the ECS and scopolamine. In the experiments presented in this study, interesting conclusions regarding this relationship were observed. The modulating effect of the compounds affecting the functioning of the ECS on the amnestic effects induced by scopolamine, confirm the previously described relationship between the ECS and the cholinergic system, and is consistent with the available literature data [38,39,70].

There is also literature data indicating that nicotine itself reverses scopolamine-dependent impairment of memory processes, which in turn suggests that the mechanism of action of scopolamine is related not only to M receptors but also to the other neurotransmitter systems [23,33,71].

As mentioned before, the ECS exhibits the interaction with, inter alia, NMDA receptors. Considering the fact that NMDA receptors are also involved in the amnestic effects of scopolamine [58,70], it is likely that the reversal of scopolamine amnesia by administration of CB1 receptor antagonists, demonstrated in the presented study, may be just a result of the interaction between CB1 and NMDA receptors. It is also intriguing that CB1 receptors and cholinergic receptors N and M are distributed together in various structures of the brain, i.e., the hippocampus and the amygdala, where we can also see the existence of functional interactions between these neurotransmitter systems [72]. At the same time, there are few scientific studies on the correlation between CB2-receptors and M cholinergic receptors. Perhaps this is due to the fact that the very functioning of CB2-receptors has been described only by relatively few researchers. The results of our studies, which show that both agonists and antagonists of CB2-receptors inhibit the amnestic effects of scopolamine, and also enhance the procognitive effects of nicotine, clearly suggest the existence of interactions between CB2 and cholinergic receptors.

Due to the extremely diverse involvement of the ECS in all life functions of the body, largely dependent on the distribution of CB1- and CB2-receptors, their far-reaching interactions with many neurotransmitter systems, interaction with the other systems in the modification of a number of physiological processes, it seems extremely important to achieve an in-depth knowledge of the specifics of the operations of the ECS.

Moreover, the results of the experiments presented here indicate the possibility of using selective ligands of CB2-receptors as compounds potentially improving cognitive processes. It also seems that the use of selective CB2-receptor ligands is more advantageous than selective CB1 receptor ligands due to their affinity for microglial cells affected by neurodegenerative changes, and also because they are largely devoid of psychotropic effects i.e., dizziness, hallucinations, and drowsiness, which may occur after administration of CB1 receptor ligands [73].

The results described in the present study indicate, first of all, the possibility of using compounds that modulate the activity of the ECS indirectly, in the treatment of neurodegenerative diseases. The outcomes of the experiments conducted are extremely promising and give hope for finding new, innovative lines of pharmacotherapy, which, by indirectly affecting several involved neurotransmitter systems, will ultimately lead to desired memory improvement. The results presented in this work indicate the possibility of combining CB2-receptor ligands with nicotine in a dose that does not affect memory on its own, in order to obtain a pharmacological effect in the pro-cognitive direction, while reducing the side effects associated with the use of nicotine; this drug therapy could find application in the treatment of diseases involving memory deficits, e.g., AD.

The results of the experiments presented in this study are promising; however, more research work is needed to fully understand the mechanisms by which the ECS is involved in modulating cognitive processes.

## 4. Materials and Methods

### 4.1. Animals

The experiments were carried out on naive male Swiss mice (Farm of Laboratory Animals, Warszawa, Poland) weighing 20–30 g; 4 weeks of age. The animals were maintained under standard laboratory conditions (12-h light/dark cycle, room temperature at 21 ± 1 °C) with free access to tap water and laboratory feeding (Agropol, Motycz, Poland) in their home cages, and adapted to the laboratory conditions for at least 1 week. Each experimental group consisted of 8–10 animals. All behavioral experiments were performed between 8:00 and 15:00 and were conducted in accordance with the National Institute of Health Guidelines for the Care and Use of Laboratory Animals and the European Community Council Directive for the Care and Use of laboratory animals of 22 September 2010 (2010/63/EU). 

### 4.2. Drugs

The cholinergic receptor ligands: Nicotine (0.05, 0.1 mg/kg) (Tocris, Minneapolis, MN, USA), a cholinergic nicotinic receptor agonist, Scopolamine (1 mg/kg) (Tocris, USA), a cholinergic muscarinic receptor antagonist,

The CB2-receptor ligands: JWH 133 (0.25, 0.5, 1 mg/kg) (Tocris, USA), a potent selective CB2-receptor agonist, AM 630 (0.25, 0.5, 1 mg/kg) (Tocris, USA), a competitive CB2-receptor antagonist.

Cholinergic receptor ligands were dissolved in a saline solution (0.9% NaCl). In turn, CB2-receptor ligands were suspended in a 1% solution of Tween 80 (Sigma, St. Louis, MO, USA) in a saline solution. Nicotine was administered subcutaneously (s.c.), and scopolamine and CB2-receptor ligands were administered intraperitoneally (i.p.) at a volume of 10 mL/kg. Fresh drug solutions were prepared on each day of experimentation. Control groups received injections of saline with Tween 80 at the same volume and by the same route of administration.

Experimental doses of cholinergic and CB2-receptor ligands used for behavioral experiments and procedures were chosen accordingly to those frequently used in literature and our previous experiences [6,7,12,74].

### 4.3. Experimental Procedures

Memory-related effects were measured by the passive avoidance (PA) test, commonly used to examine different stages of memory. In the PA test, mice learn to avoid an unpleasant stimulus by inhibiting locomotion and exploration, which is why it is also called the inhibitory avoidance test.

The apparatus of PA is divided into bright (10 × 12 × 15 cm^3^) and dark compartments (14 × 18 × 15 cm^3^). The lighted chamber was illuminated by a fluorescent light (8 W) and was connected to the dark chamber which was equipped with an electric grid floor. The entrance of animals into the dark box was punished by an electric foot shock (0.2 mA for 2 s). 

The measured parameter is a latency time (TL) to enter into the black compartment. On the first day of training (pre-test), mice were placed individually into the light compartment and allowed exploring the light box. After 30 s, the guillotine door was raised to allow the mice to enter the dark compartment. When the mice entered to dark compartment, the guillotine door was closed and an electric foot-shock (0.2 mA) lasting 2 s was delivered immediately to the animal via the grid floor. The latency time for entering the dark compartment was recorded (TL1). 24 h later, in the subsequent trial (test), the same mice were again placed individually in the light compartment of the PA apparatus. After a 30 s adaptation period in the light (safe) chamber, the door between the compartments was raised and the time taken to re-enter the dark compartment was recorded (TL2).

Depending on the procedure used, the PA test allows for the examination of different stages of memory (acquisition, consolidation), according to the drug treatment duration. Drug administration before the first trial (pretest) should interfere with the acquisition of information, while drug administration immediately after the first trial (after the pre-test) should exert an effect on the process of consolidation [75].

For the memory-related responses, the changes in PA performance were expressed as the difference between pre-test and test latencies and was taken as an latency index (LI). 

LI was calculated for each animal and reported as the ratio: (1)$$\mathrm{LI}=\frac{\left(TL2-TL1\right)}{TL1}$$
where: TL1 = the time taken to enter the dark compartment during the pre-test; TL2 = the time taken to re-enter the dark compartment during the test.

### 4.4. Treatment

For the acquisition of memory, cholinergic ligands (nicotine (0.05 and 0.1 mg/kg, s.c.) or scopolamine (1 mg/kg, i.p.), CB2-receptor ligands (JWH 133 (0.25–1 mg.kg, i.p) or AM 630 (0.25–1, i.p.) or vehicle, for the control group, were administered 30 min before the first trial, and mice were re-tested after 24 h (long-term memory). 

For the consolidation of memory, cholinergic ligands (nicotine (0.05 and 0.1 mg/kg, s.c.) or scopolamine (1 mg/kg, i.p.), CB2-receptor ligands (JWH 133 (0.25–1 mg.kg, i.p) or AM 630 (0.25–1, i.p.) or vehicle, for the control group, were injected immediately after the first trial, and mice after 24 h (Table 1 and Table 2). 

Based on these pilot experiments in the PA test, we have chosen for the next experiments:the non-effective and effective dose of nicotine (0.05 mg/kg and 0.1 mg/kg, respectively)the effective dose of scopolamine (1 mg/kg)non-effective dose of JWH 133(0.25 mg/kg) and AM 630 (0.25 mg/kg).

After that, we evaluated the influence of CB2-receptor ligands, agonist and antagonist, on the memory-related responses induced by cholinergic receptor ligands in the PA task. 

Non-effective JWH 133 (0.25 mg/kg, i.p.) or vehicle was administered acutely 15 min before an acute injection of nicotine (0.05 or 0.1 mg/kg, s.c.), or scopolamine (1 mg/kg, i.p) or vehicle. Similarly, non-effective AM 630 (0.25 mg/kg, i.p.) or vehicle was administered acutely 15 min before an acute injection of nicotine (0.05 or 0.1 mg/kg, s.c.), or scopolamine (1 mg/kg; i.p.) or vehicle. The acquisition or consolidation of long-term memory was tested afterwards in the same scheme described above and presented in Table 3 and Table 4.

### 4.5. Statistical Analysis

The statistical analysis was performed using one-way analysis of variance (ANOVA) or two-way ANOVA for the factors of pretreatment (JWH 133 or AM 630), treatment (nicotine or scopolamine), and pretreatment/treatment interactions for the memory-related responses. Post hoc comparison of means was carried out with the Tukey’s test (for one-way ANOVA) or with the Bonferroni’s test (for two-way ANOVA) for multiple comparisons, when appropriate. The data were considered statistically significant at a confidence limit of *p* < 0.05. ANOVA analysis with Tukey’s or Bonferroni’s post hoc tests was performed using GraphPad Prism version 5.00 for Windows, GraphPad Software, San Diego California USA, www.graphpad.com (accessed on 25 June 2022).

For the memory-related responses, the changes in PA performance were expressed as the difference between retention and training latencies and were taken as a latency index (LI). LI was calculated for each animal and reported as the ratio: LI = TL2-TL1/TL1, where TL1 is the time taken to enter the dark compartment during the training and TL2 is the time taken to re-enter the dark compartment during the retention (Chimakurthy and Talasila 2010).

## 5. Conclusions

Our experiments show that ECS, through CB2-receptors, participates in the modulation of memory processes, especially those in which cholinergic pathways are implicated. Co-administration of CB2-receptor agonist JWH 133 (0.25 mg) or CB2-receptor antagonist AM 630 (0.25 mg/kg) with the non-effective dose of nicotine (0.05 mg/kg) enhanced cognition in the PA test in mice; however, an acute injection of JWH 133 (0.25 mg/kg) or AM 630 (0.25 mg/kg) had no influence on memory enhancement induced by the effective dose of nicotine (0.1 mg/kg). In turn, co-administration of JWH 133 (0.25 mg) or AM 630 (0.25 mg/kg) with the effective dose of scopolamine (1 mg/kg) attenuated the scopolamine-induced memory impairment in the PA test in mice. 

The results of the studies presented in this paper, which show that both agonists and antagonists of CB2-receptors inhibit the amnestic effects of scopolamine, and also enhance the pro-cognitive effects of nicotine, clearly suggest the existence of interactions between CB2 and cholinergic receptors. Naturally, such interactions may take place at the receptor level (CB2 and cholinergic receptors, but not necessarily synergism or antagonism). These results may be a consequence of the more complex mechanisms of action of the tested substances; thus, further research is necessary. In the future, our findings could be helpful in further research concerning the effective pharmacotherapy in diseases that are associated with cognitive impairments, especially those referred to as cholinergic system dysfunction, such as AD.

## Figures and Tables

**Figure 1 molecules-27-04252-f001:**
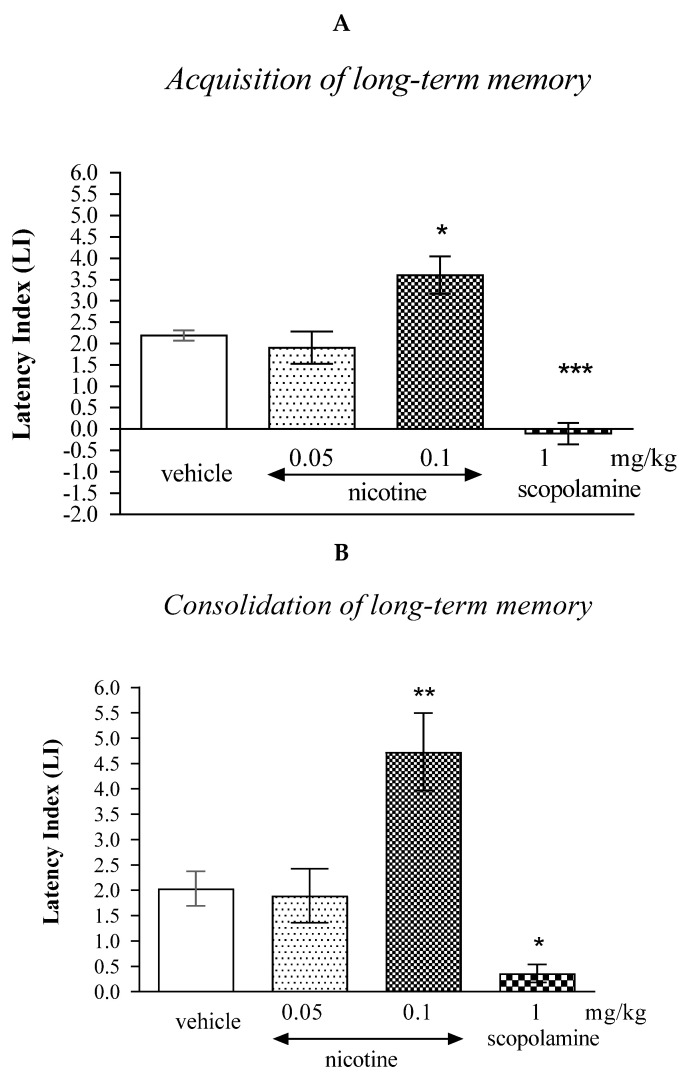
The influence of an acute administration of cholinergic receptor ligands on the cognition-related responses, expressed as latency index (LI), during the acquisition (**A**) or consolidation of memory using the PA test in mice (**B**). Nicotine (0.05 or 0.1 mg/kg), scopolamine (1 mg/kg) or vehicle were administered 30 min before the first trial (acquisition of memory) or immediately after the first trial (consolidation of memory). The second trial was conducted 24 h after the first one; n = 8–10; the means ± SEM; * *p* < 0.05; ** *p* < 0.01; *** *p* < 0.001 vs. vehicle-treated group; Tukey’s test.

**Figure 2 molecules-27-04252-f002:**
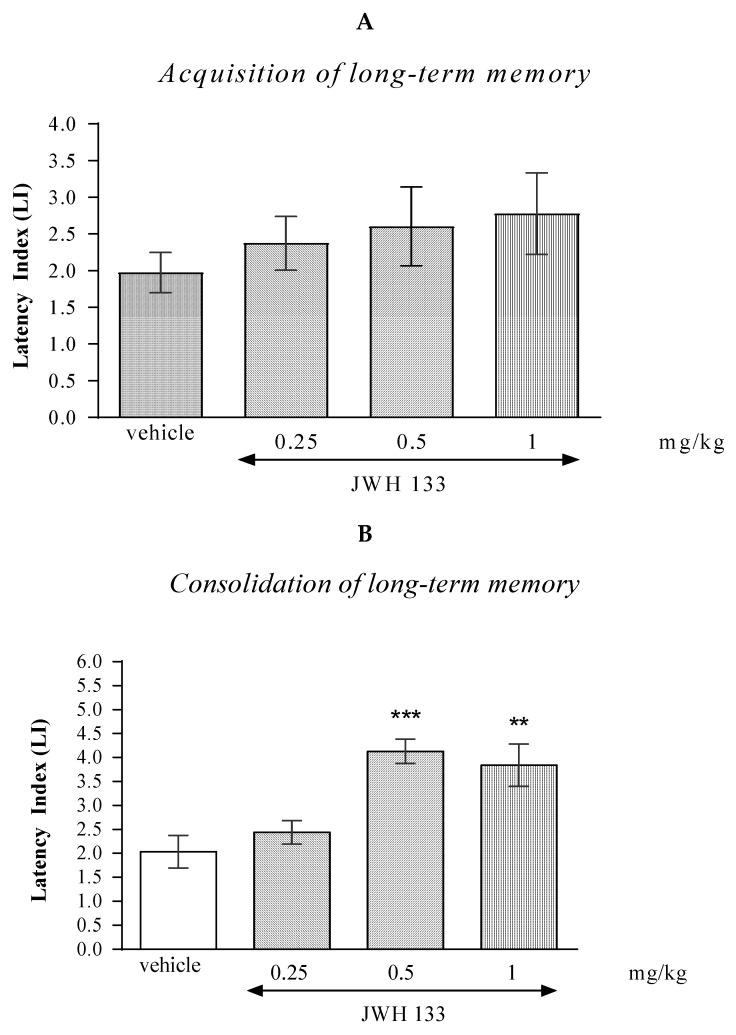
The influence of an acute administration of JWH 133 on the cognition-related responses, expressed as latency index (LI), during the acquisition (**A**) and consolidation (**B**) of memory using the PA test in mice. JWH 133 (0.25, 0.5 or 1 mg/kg) or vehicle were administered 30 min before the first trial (acquisition of memory) or immediately after the first trial (consolidation of memory). The second trial was conducted 24 h after the first one; *n* = 8–10; the means ± SEM; ** *p* < 0.01; *** *p* < 0.001 vs. vehicle-treated group; Tukey’s test.

**Figure 3 molecules-27-04252-f003:**
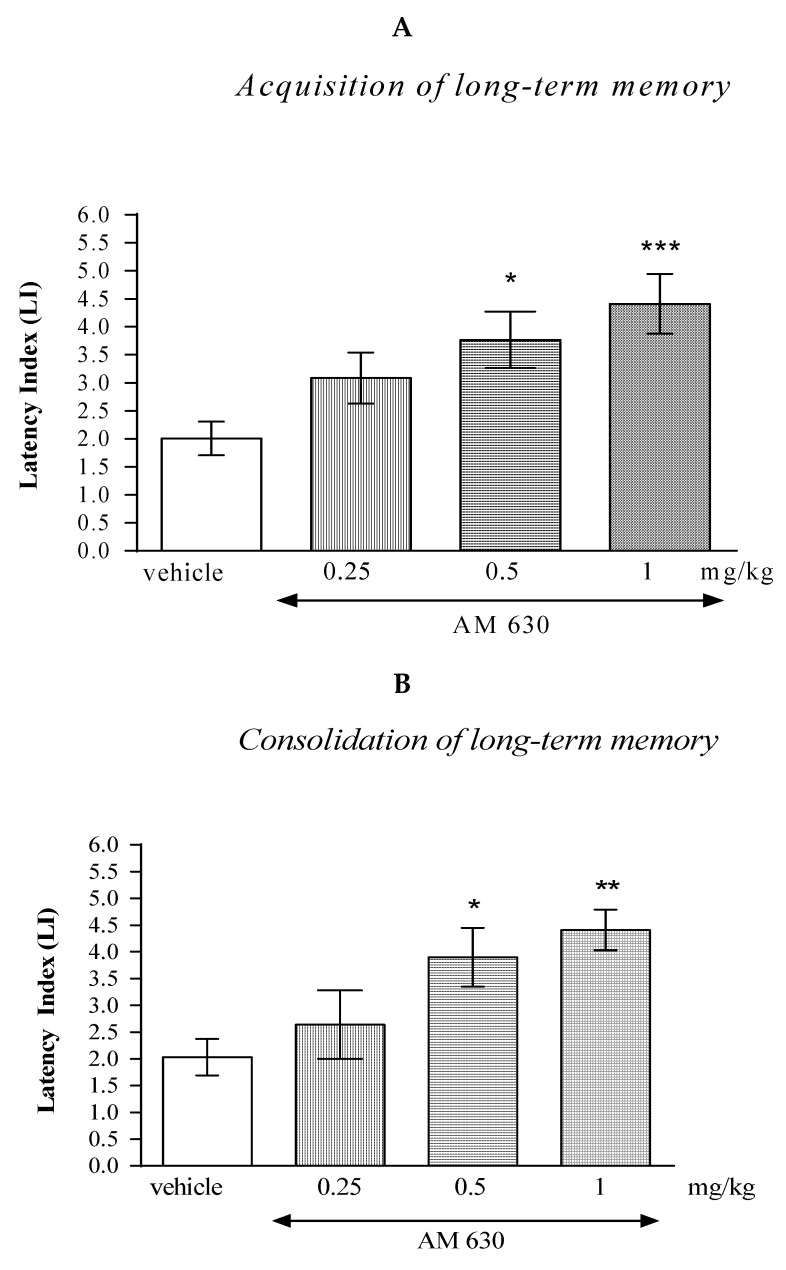
The influence of an acute administration of AM 630 on the cognition-related responses, expressed as latency index (LI), during the acquisition (**A**) and consolidation (**B**) of memory using the PA test in mice. AM 630 (0.25, 0.5 or 1 mg/kg) or vehicle were administered 30 min before the first trial (acquisition of memory) or immediately after the first trial (consolidation of memory). The second trial was conducted 24 h after the first one; *n* = 8–10; the means ± SEM; * *p* < 0.05; ** *p* < 0.01; *** *p* < 0.001 vs. vehicle-treated group; Tukey’s test.

**Figure 4 molecules-27-04252-f004:**
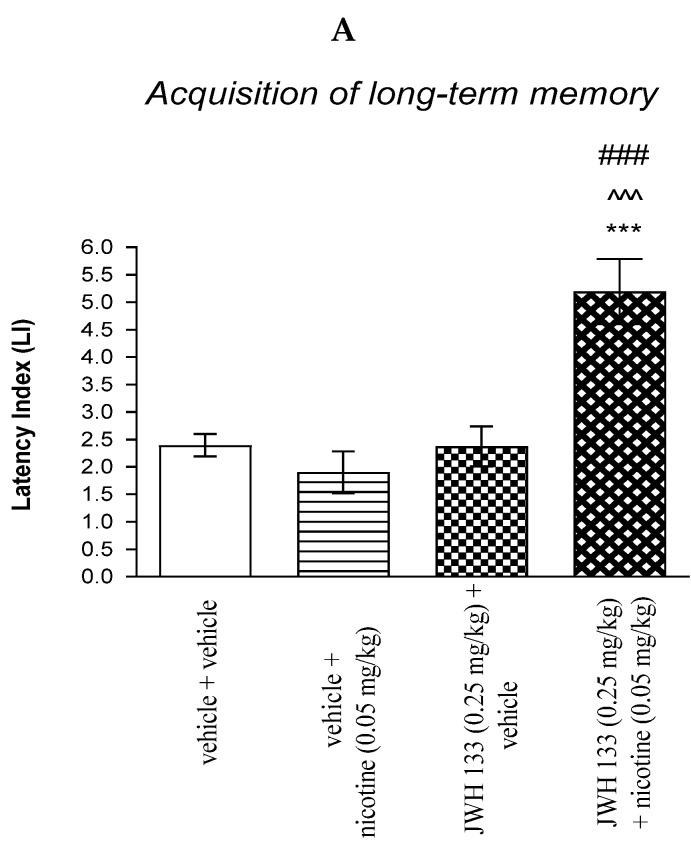
The influence of an acute co-administration of CB2 receptor agonist and non-effective dose of nicotine on the cognition-related responses expressed as latency index (LI) during the acquisition (**A**) and consolidation (**B**) memory using the PA test in mice. *** *p* < 0.001 vs. vehicle/vehicle-treated mice; ^^^ *p* < 0.001 vs. vehicle/nicotine (0.05 mg/kg)-treated mice; ### *p* < 0.001 vs. JWH 133 (0.25 mg/kg)/vehicle-treated mice; ** *p* < 0.01 vs. vehicle/vehicle-treated mice; # *p* < 0.05 vs. JWH 133 (0.25 mg/kg)/vehicle-treated mice.

**Figure 5 molecules-27-04252-f005:**
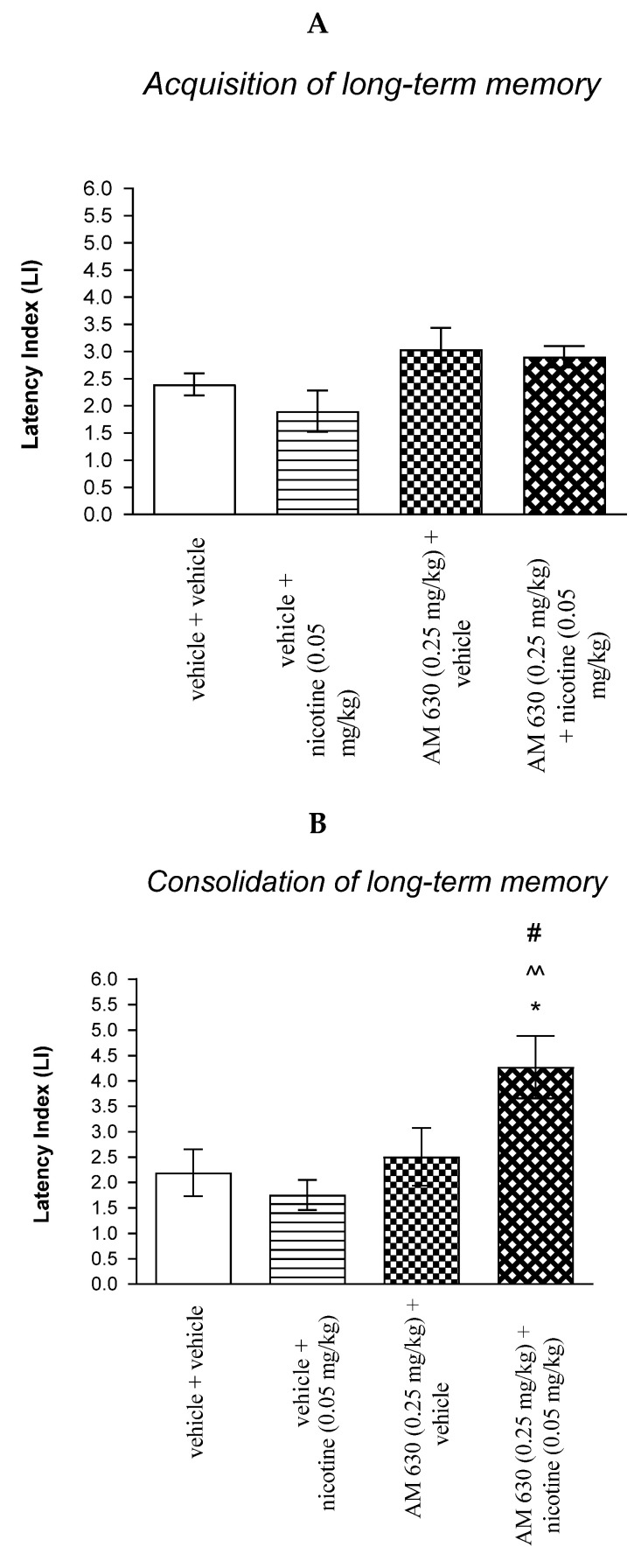
The influence of an acute co-administration of CB2 receptor antagonist and non-effective dose of nicotine on the cognition-related responses expressed as latency index (LI) during the acquisition (**A**) and consolidation (**B**) memory using the PA test in mice. * *p* < 0.05 vs. vehicle/vehicle-treated mice; ^^ *p* < 0.01 vs. vehicle/nicotine (0.05 mg/kg)-treated mice; # *p* < 0.05 vs. AM 630 (0.25 mg/kg)/vehicle-treated mice.

**Figure 6 molecules-27-04252-f006:**
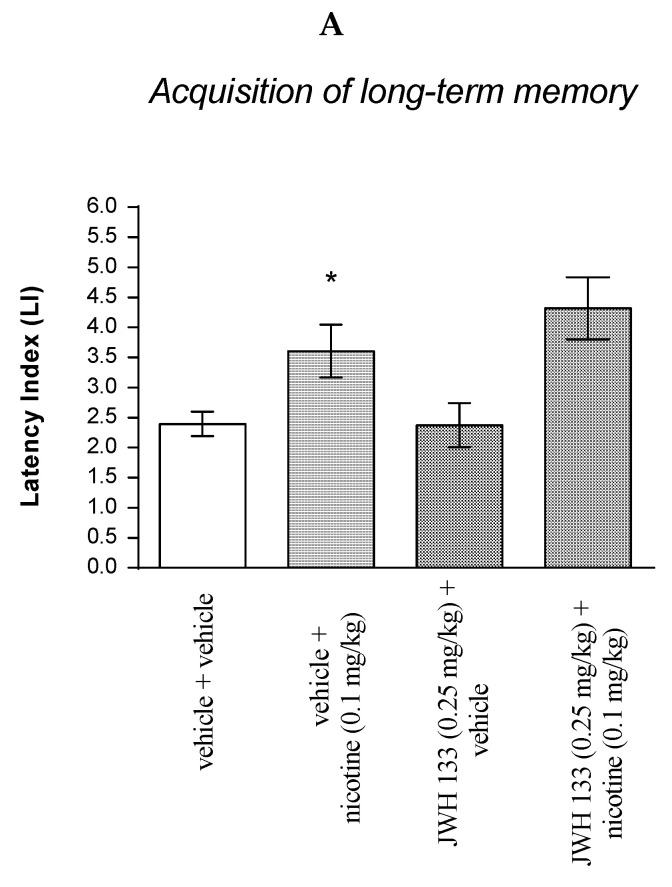
The influence of an acute co-administration of CB2 receptor agonist and effective (pro-cognitive) dose of nicotine on the cognition-related responses expressed as latency index (LI) during the acquisition (**A**) and consolidation (**B**) of memory using the PA test in mice. * *p* < 0.05 vs. vehicle/vehicle-treated mice; * *p* < 0.05 vs. vehicle/vehicle-treated mice.

**Figure 7 molecules-27-04252-f007:**
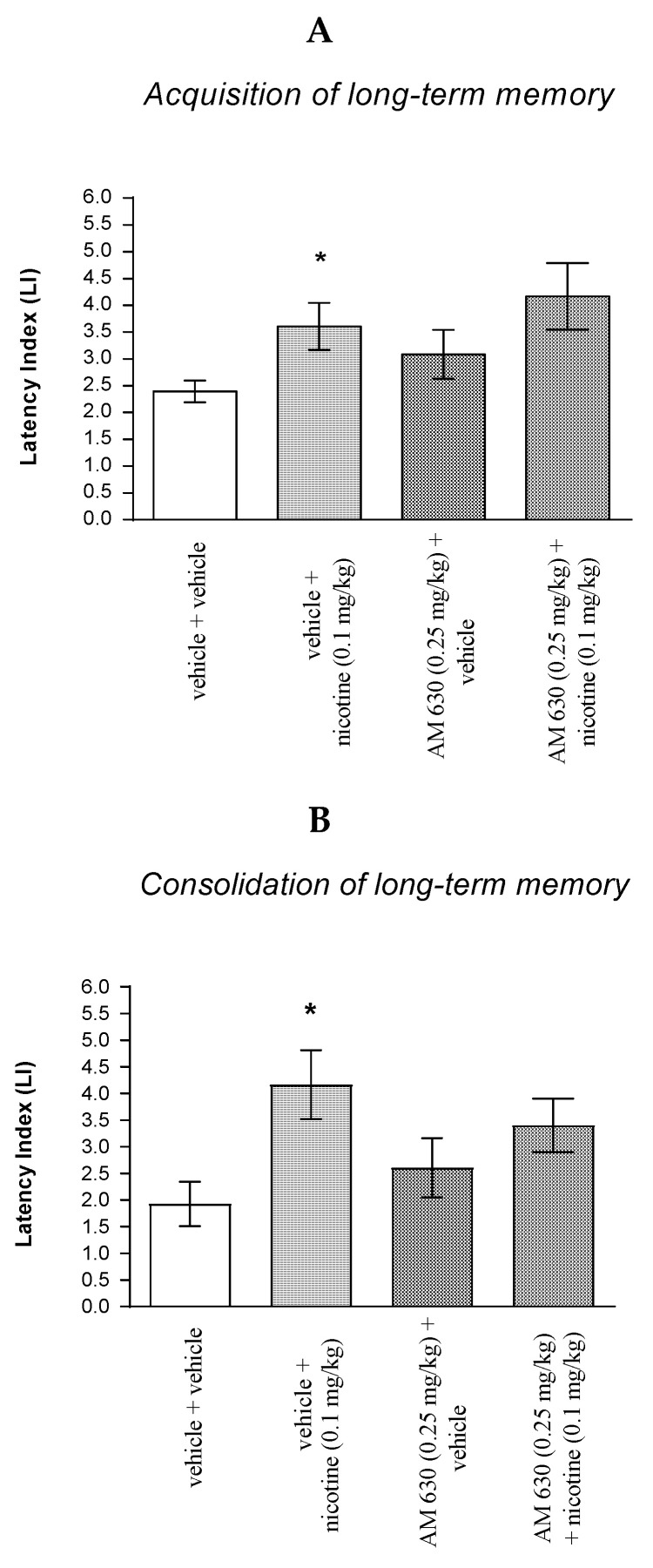
The influence of an acute co-administration of CB2-receptor antagonist and effective (pro-cognitive) dose of nicotine on the cognition-related responses expressed as latency index (LI) during the acquisition (**A**) and consolidation (**B**) of memory using the PA test in mice. * *p* < 0.05 vs. vehicle/vehicle-treated mice.

**Figure 8 molecules-27-04252-f008:**
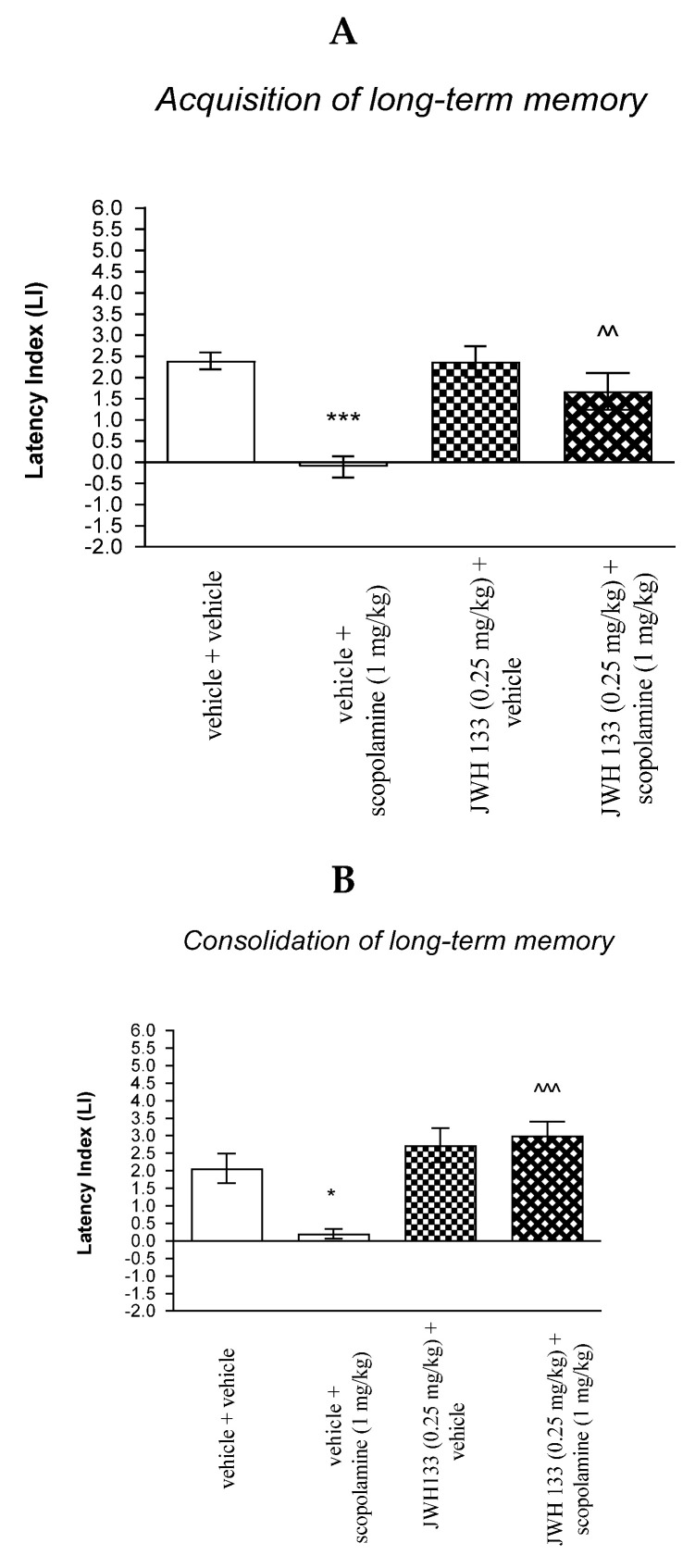
The influence of an acute co-administration of CB2-Receptor agonist and effective (amnestic) dose of scopolamine on the cognition-related responses expressed as latency index (LI) during the acquisition (**A**) and consolidation (**B**) memory using the PA test in mice. * *p* < 0.05 vs. vehicle/vehicle-treated mice; *** *p* < 0.001 vs. vehicle/vehicle-treated mice; ^^ *p* < 0.01 vs. vehicle/scopolamine (1 mg/kg)-treated mice; ^^^ *p* < 0.001 vs. vehicle/scopolamine (1 mg/kg)-treated mice.

**Figure 9 molecules-27-04252-f009:**
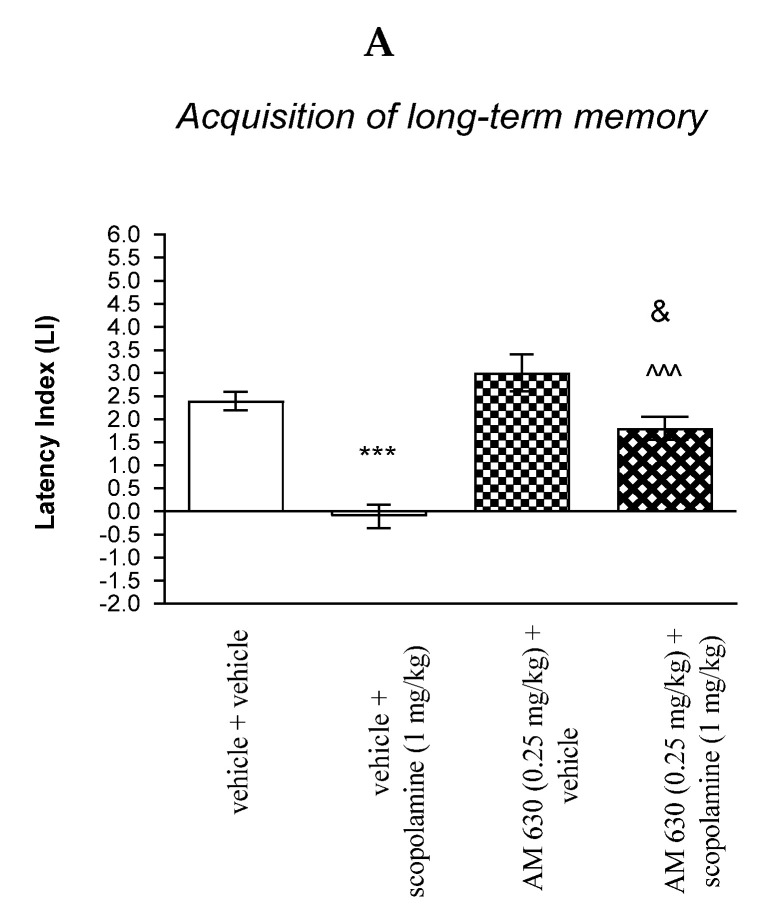
The influence of an acute co-administration of CB2-receptor antagonist and effective (amnestic) dose of scopolamine on the cognition-related responses expressed as latency index (LI) during the acquisition (**A**) and consolidation (**B**) of memory using the PA test in mice. *** *p* < 0.001 vs. vehicle/vehicle-treated mice; ^^^ *p* < 0.001 vs. vehicle/scopolamine (1 mg/kg)-treated mice; ^&^
*p* < 0.05 vs. AM 630 (0.25 mg/kg)/vehicle-treated mice.

**Table 1 molecules-27-04252-t001:** The scheme of cholinergic and CB2-receptor ligands or vehicle administration during the assessment of long-term memory acquisition in the PA test in mice.

Acquisition of Memory					
PA Test	Drug Administration	Interval	TL1	Interval	TL2
**Long-term memory**	Nicotine (0.05 or 0.1 mg/kg)	30 min	+	24 h	+
	Scopolamine (1 mg/kg)	30 min	+	24 h	+
	JWH 133 (0.25; 0.5; 1 mg/kg)	30 min	+	24 h	+
	AM 630 (0.25; 0.5; 1 mg/kg)	30 min	+	24 h	+
	Vehicle (for the control group)	30 min	+	24 h	+

**Table 2 molecules-27-04252-t002:** The scheme of cholinergic and CB2-receptor ligands or vehicle administration during the assessment of long-term memory consolidation in the PA test in mice.

Consolidation of Memory					
PA Test	TL1	Interval	Drug Administration	Interval	TL2
**Long-term memory**	+	0 min	Nicotine (0.05 or 0.1 mg/kg)	24 h	+
	+	0 min	Scopolamine (1 mg/kg)	24 h	+
	+	0 min	JWH 133 (0.25; 0.5; 1 mg/kg)	24 h	+
	+	0 min	AM 630 (0.25; 0.5; 1 mg/kg)	24 h	+
	+	0 min	Vehicle (for the control group)	24 h	+

**Table 3 molecules-27-04252-t003:** The scheme of CB2 and cholinergic receptor ligands co-administration during the assessment of long-term memory acquisition in the PA test in mice.

Acquisition of Memory							
PA Test	Drug Administration	Interval	Drug Administration	Interval	TL1	Interval	TL2
**Long-term memory**	JWH 133 (0.25 mg/kg)	15 min	nicotine (0.05 mg/kg) or nicotine (0.1 mg/kg) or scopolamine (1 mg/kg) or vehicle	15 min	+	24 h	+
	AM 630 (0.25 mg/kg)	15 min	nicotine (0.05 mg/kg) or nicotine (0.1 mg/kg) or scopolamine (1 mg/kg) or vehicle	15 min	+	24 h	+
	vehicle (control group)	15 min	nicotine (0.05 mg/kg) or nicotine (0.1 mg/kg) or scopolamine (1 mg/kg) or vehicle	15 min	+	24 h	+

**Table 4 molecules-27-04252-t004:** The scheme of CB2 and cholinergic receptor ligands co-administration during the assessment of long-term memory consolidation in the PA test in mice.

Consolidation of Memory							
PA Test	TL1	Interval	Drug Administration	Interval	Drug Administration	Interval	TL2
**Long-term memory**	+	0 min	JWH 133 (0.25 mg/kg)	15 min	nicotine (0.05 mg/kg) or nicotine (0.1 mg/kg) or scopolamine (1 mg/kg) or vehicle	24 h	+
	+	0 min	AM 630 (0.25 mg/kg)	15 min	nicotine (0.05 mg/kg) or nicotine (0.1 mg/kg) or scopolamine (1 mg/kg) or vehicle	24 h	+
	+	0 min	vehicle (control group)	15 min	nicotine (0.05 mg/kg) or nicotine (0.1 mg/kg) or scopolamine (1 mg/kg) or vehicle	24 h	+

## Data Availability

Data is contained within the article.

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
