# Peer review of "The Influence of CB2-Receptor Ligands on the Memory-Related Responses in Connection with Cholinergic Pathways in Mice in the Passive Avoidance Test"

_molecules, 2022, doi:10.3390/molecules27134252_

Round 1
Reviewer 1 Report
This is interesting study by Marta et al showing cannabinoid ligands effect on memory related response. Overall study is based on pharmacological compounds that are highly dose-sensitive. Authors used CB2 agonist JWH133 and CB2 inverse agaonist AM630, both have shown good memory retention at higher dose which may indicate towards non-intended effect of these compounds. This not necessarily showing CB2-dependent effect. This is also confirmed by non-effective dose of nicotine with combination of JWH133 or AM630 enhances memory. Does it mean to synergistic effect? I am more puzzled to see scopolamine has no effect on memory when given in combination with non-effective low dosages of JWH133 or AM630.
There are several lacuna in the study
1. the study relies on single behavior function, which may not depict the complex memory mechanism. Authors need to use other tools such Morris Water Maze, Novel object recognition or Barne's maze in combination with passive avoidance test, specially when results are ambiguous.
2. No ligands is specific to only single receptor. Authors need to use Cb1 agonist and antagonist too. The results here pointing towards involvement of cB1 than CB2.
3. discussion is long and lacks interpretation of study results. There is not clear conclusion of study.
Author Response
Answer for Reviewer 1:
The study relies on single behavior function, which may not depict the complex memory mechanism. Authors need to use other tools such Morris Water Maze, Novel object recognition or Barne's maze in combination with passive avoidance test, specially when results are ambiguous.
Passive avoidance as an animal model of memory is hard to classify in terms of the fundamental types of memory. However, passive avoidance is the behavioral procedure of choice in many studies of learning and memory, probably because it requires little special training of the subjects, and because results are available quickly. Unfortunately, comparisons across studies are made difficult by the fact that very different procedures are used in studies of passive avoidance in different laboratories. Different tasks are employed (e.g., step-through, step-down, etc.), and different shock parameters are used (varying intensities and numbers of escapable or inescapable shocks) [Willson and Cook, 1994].
According to Venault et al. [1986] the step-through passive avoidance task may be recognized as a measure of short- and long-term memory. In our experiments (present and in the past) we used the procedure of PA task, which is commonly approved in the assessment of memory-related responses [Allami et al., 2012; Borowicz et al., 1995; Chimakurthy and Talasila, 2010; Hiramatsu et al., 1998; Javadi-Paydar et al., 2012] and allows us to obtain reproducible and reliable results.
PA task is commonly used to investigate emotional learning and memory processes in rodents (Gold, 1986; McGaugh and Roozendaal, 2009). PA task combines Pavlovian contextual fear conditioning with the expression of an instrumental response, the avoidance of entering a particular (punished) area of the training context (Ögren and Stiedl, 2010). In the PA paradigm, alterations in the response latency have been thought to reflect the degree of memory, however, the emotionality (fear and/or anxiety) of animals can presumably affect the avoidance behavior [Nishimura et al., 1989].
In the case of continuing the experiments with the use of the tested compounds and their various combinations, we will plan the experiments using other, suggested by the Reviewer, tests for additional assessment of cognitive behavior which could make the results more precise. At the moment, our unit does not have the above-mentioned devices at its disposal, but we are in the process of planning their purchases and applying for financing from external sources.
No ligands is specific to only single receptor. Authors need to use CB1 agonist and antagonist too. The results here pointing towards involvement of CB1 than CB2.
Research on the effect of CB1 receptor ligands was also carried out by us in a similar scheme. We used an agonist of CB1 receptor – oleamide and an antagonist of CB1 receptor – AM 251. The publication is in preparation.
When describing the research results, we separated the CB1 and CB2 receptors according to the number of results. First, we wanted to publish the effects of CB2 ligands because this type of receptor has not yet been so thoroughly studied and there are few results in the scientific literature.
Discussion is long and lacks interpretation of study results. There is not clear conclusion of study.
As the Reviewer suggested the discussion and conclusion has been corrected and shortened.
Concerning the question Reviewer: “ This is interesting study by Marta et al showing cannabinoid ligands effect on memory related response. Overall study is based on pharmacological compounds that are highly dose-sensitive. Authors used CB2 agonist JWH133 and CB2 inverse agonist AM630, both have shown good memory retention at higher dose which may indicate towards non-intended effect of these compounds. This not necessarily showing CB2-dependent effect. This is also confirmed by non-effective dose of nicotine with combination of JWH133 or AM630 enhances memory. Does it mean to synergistic effect? I am more puzzled to see scopolamine has no effect on memory when given in combination with non-effective low dosages of JWH133 or AM630.
The results of the studies presented in this paper, which show that both agonists and antagonists of CB2 receptors inhibit the amnestic effects of scopolamine, and also enhance the procognitive effects of nicotine, clearly suggest the existence of interactions between CB2 and cholinergic receptors. Of course, such interactions may take place at the receptor level (CB 2 and nicotinic or muscarinic receptors, but not necessarily synergism or antagonism). I agree with the Reviewer that our results may be a consequence of the more complex mechanisms of action of the tested substances which has been discussed in our MS.
Additionally, there are few scientific studies on the correlation between CB2 receptors and cholinergic receptors. However, these are only suggestions and further research is necessary.
Additionally, a complete revision of the text of manuscript has been done and all English typos have been corrected.
All changes made in the manuscript are marked in red font.
Reviewer 2 Report
The authors presented an interesting study on nootropic effects of CB2 ligands. Study protocol is well designed, and results are clearly presented, but some issues have to be reworked.
1. PA test is not clearly described. Parameter TL1/TL1 in equation for LI is confusing (same/same = 1)
2. Declare the age of animals used in the study since it can affect neurocognitive processes.
3. Why were CB2 ligands dissolved in the Tween 80? Can they be dissolved in a more inert solvent (e.g. oil)?
4. When mentioning ligand of the receptor, please declare particular subtype of the receptor that ligand refers to.
5. The discussion is too long. Avoid detailed elaboration of topics being not directly related to the subject (e.g. inflammation)
Discussion needs rewording and improvement of english and style (eg. "scientist have proven..."; "paperwork".......)
List of abbreviations is unnecessary. Write the full name after first mentioning in the text.
Author Response
Answer for Reviewer 2:
PA test is not clearly described. Parameter TL1/TL1 in equation for LI is confusing (same/same = 1)
I agree with the Reviewer that the PA procedure should be described more clearly. We have made a certain correction.
Declare the age of animals used in the study since it can affect neurocognitive processes.
It seems that information-handling processes are disturbed in old animals, although hypotheses diverge as to which subprocess is affected. One hypothesis is that old animals are less effective at consolidating what has been learned (Weiskrantz & Warrington, 1975), and another is that the disturbing factor is great perseverance, i.e., excessive resistance to extinction of what has been learned and to the integration of new information (Lapsley & Enright, 1983). Greater agreement exists, however, with respect to neuroanatomic and neurofunctional, age-typical changes in the hippocampus, which are directly associated with learning deficits (Landfield, 1988; Winocur, 1988)
As the Reviewer suggested the age of animals used in the study has been added in the Material an Method Section .
Why were CB2 ligands dissolved in the Tween 80? Can they be dissolved in a more inert solvent (e.g. oil)?
Cannabinoid compounds are lipophilic thus they dissolve in oily solvents. The recommendations of the Tocris company, where we purchased the AM 630 and JWH 133, are to dissolve cannabinoid ligands in DMSO (AM 630) or DMSO or ethanol (JWH 133).
We chose Tween 80 because it has similar properties to DMSO [Castro et al, 1995] and has been widely used as a solvent for pharmacological experiments [Li and Kim, 2016; Kruk-Slomka et al., 2016; 2017; Ratano et al., 2017]. What is of interest, the literature data described an experiment examined the effects of using a 5% alcohol solution as a solvent for cannabinoid agonists (CP 55,940), in comparison to the more inert detergent Tween 80. The results suggest 1 ml/kg of a 5% alcohol solution has significant behavioral effects on its own [Stanley-Cary, Harris et al., 2002].
When mentioning ligand of the receptor, please declare particular subtype of the receptor that ligand refers to.
I agree with the Reviewer that the subtype of CB receptor should be marked. We have made a certain correction.
The discussion is too long. Avoid detailed elaboration of topics being not directly related to the subject (e.g. inflammation)
Discussion needs rewording and improvement of english and style (eg. "scientist have proven..."; "paperwork".......)
As the Reviewer suggested, the Discussion section has been re-viewed, shortened and re-written.
List of abbreviations is unnecessary. Write the full name after first mentioning in the text.
As the Reviewer suggested the list of abbreviations has been removed from manuscript.
Additionally, a complete revision of the text of manuscript has been done and all English typos have been corrected.
All changes made in the manuscript are marked in red font.